# Hospital-Acquired Serum Ionized Calcium Derangements and Their Associations with In-Hospital Mortality

**DOI:** 10.3390/medicines7110070

**Published:** 2020-11-19

**Authors:** Charat Thongprayoon, Panupong Hansrivijit, Tananchai Petnak, Michael A. Mao, Tarun Bathini, Saraschandra Vallabhajosyula, Ploypin Lertjitbanjong, Fawad Qureshi, Stephen B. Erickson, Wisit Cheungpasitporn

**Affiliations:** 1Division of Nephrology and Hypertension, Department of Medicine, Mayo Clinic, Rochester, MN 55905, USA; Qureshi.Fawad@mayo.edu (F.Q.); Erickson.Stephen@mayo.edu (S.B.E.); 2Department of Internal Medicine, University of Pittsburgh Medical Center Pinnacle, Harrisburg, PA 17101, USA; hansrivijitp@upmc.edu; 3Division of Pulmonary and Pulmonary Critical Care Medicine, Faculty of Medicine, Ramathibodi Hospital, Mahidol University, Bangkok 10400, Thailand; petnak@yahoo.com; 4Division of Pulmonary and Critical Care Medicine, Department of Medicine, Mayo Clinic, Rochester, MN 55905, USA; 5Division of Nephrology and Hypertension, Mayo Clinic, Jacksonville, FL 32224, USA; mao.michael@mayo.edu; 6Department of Internal Medicine, University of Arizona, Tucson, AZ 85721, USA; tarunjacobb@gmail.com; 7Section of Interventional Cardiology, Division of Cardiovascular Medicine, Department of Medicine, Emory University School of Medicine, Atlanta, GA 30322, USA; saraschandra21@gmail.com; 8Division of Pulmonary, Critical Care and Sleep Medicine, University of Tennessee Health Science Center, Memphis, TN 38163, USA; ploypinlert@gmail.com

**Keywords:** hypercalcemia, hypocalcemia, calcium, electrolytes, mortality, hospitalization

## Abstract

**Background**: The objective of this study was to report the incidence of in-hospital serum ionized calcium derangement and its impact on mortality. **Methods:** We included 12,599 non-dialytic adult patients hospitalized at a tertiary medical center from January 2009 to December 2013 with normal serum ionized calcium at admission and at least 2 in-hospital serum ionized calcium values. Using serum ionized calcium of 4.60–5.40 mg/dL as the normal reference range, in-hospital serum ionized calcium levels were categorized based on the presence of hypocalcemia and hypercalcemia in hospital. We performed logistic regression to assess the relationship of in-hospital serum ionized calcium derangement with mortality. **Results:** Fifty-four percent of patients developed new serum ionized calcium derangements: 42% had in-hospital hypocalcemia only, 4% had in-hospital hypercalcemia only, and 8% had both in-hospital hypocalcemia and hypercalcemia. In-hospital hypocalcemia only (OR 1.28; 95% CI 1.01–1.64), in-hospital hypercalcemia only (OR 1.64; 95% CI 1.02–2.68), and both in-hospital hypocalcemia and hypercalcemia (OR 1.73; 95% CI 1.14–2.62) were all significantly associated with increased in-hospital mortality, compared with persistently normal serum ionized calcium levels. **Conclusions:** In-hospital serum ionized calcium derangements affect more than half of hospitalized patients and are associated with increased in-hospital mortality.

## 1. Introduction

Calcium is an important cation in the human body that plays a significant role in maintaining cellular biologic function, either in the free ion or protein-bound complexes [1,2]. The majority (>99%) of total body calcium resides in the bones as hydroxyapatite, a form of calcium-phosphate complex [3]. The remaining total body calcium (approximately 10 in adults) exists outside of the skeletal system and participates in a myriad of essential functions, including neurotransmitter release, muscle contraction, and cellular signaling. Approximately 50% of circulating calcium ions are bound to protein, such as albumin or globulin, whereas the rest exists as unbound ionized (free) calcium [4,5,6,7,8]. Measuring ionized calcium is important, as Payne’s formula [6] for corrected total calcium level tends to overestimate ionized calcium, especially in patients with hypoalbuminemia [7,8]. Thus, the gold standard for evaluating calcium status is to measure ionized calcium [9,10].

Disturbances of serum calcium, hypocalcemia and hypercalcemia, are associated with increased mortality [11,12,13,14,15,16,17]. A large retrospective cohort study of the United States Department of Veterans Affairs involving 1.9 million patients suggested a U-shaped association between calcium disturbances and mortality [18]. Furthermore, long-term effects of serum calcium derangements are also possible. Both reduced and elevated admission serum ionized calcium were associated with increased 1 year mortality with a U-shape relationship [19]. While the effects of calcium disturbances at hospital admission on poor outcomes among hospitalized patients have been demonstrated [4,5,19], the incidence of in-hospital calcium disturbances and their association with mortality are currently unknown.

Therefore, the objective of this study was to investigate the incidence of in-hospital hypocalcemia and hypercalcemia, assessed by serum ionized calcium level, and their impact on in-hospital mortality among hospitalized patients.

## 2. Materials and Methods

### 2.1. Study Population

The Mayo Clinic Institutional Review Board approved this study and waived the need for informed consent as long as patients provided Minnesota research authorization, due to the minimal risk nature of this investigation.

We initially searched our institutional database to identify all patients (aged 18 years or older) who were admitted to Mayo Clinic Hospital, Rochester, Minnesota, from 2009 to 2013. Our inclusion criteria consisted of (1) patients who had normal admission serum ionized calcium levels of 4.60 to 5.40 mg/dL, and (2) patients who had at least two in-hospital serum ionized calcium values. Exclusion criteria consisted of patients who required renal replacement therapy in hospital.

### 2.2. Definition of In-Hospital Hypocalcemia and Hypercalcemia

We reviewed all in-hospital serum ionized calcium values. All serum ionized calcium values were analyzed from venous blood samples collected in serum separating tubes or serum gel tubes using the ion-selective electrode method throughout the study period. Serum ionized calcium was adjusted to pH 7.40 to account for changes in specimen pH that may occur during transport. We considered serum ionized calcium at 4.60–5.40 mg/dL as the normal range according to our hospital’s laboratory reference values [20]. We identified in-hospital hypocalcemia as having any in-hospital serum ionized calcium values <4.60 mg/dL and in-hospital hypercalcemia as having any in-hospital serum ionized calcium values >5.40 mg/dL. In-hospital serum ionized calcium levels were classified into 4 categories based on the occurrence of in-hospital hypocalcemia and hypercalcemia: (1) persistently normal serum ionized calcium, (2) in-hospital hypocalcemia only, (3) in-hospital hypercalcemia only, and (4) both in-hospital hypocalcemia and hypercalcemia.

### 2.3. Outcomes

We assessed the impact of in-hospital serum ionized calcium derangements on in-hospital mortality. We used the institutional database to determine vital status at hospital discharge.

### 2.4. Statistical Analysis

We tested the normality of continuous variables using the Shapiro-Wilk test. We presented continuous variables as mean ± standard deviation (SD) for normally distributed data, or median (interquartile rate (IQR)) for skewed data. We compared continuous variables using analysis of variance for normally-distributed data, or the Kruskal-Wallis test for skewed data. We presented categorical variables as frequency (percentage) and compared them using the chi-squared test. We obtained the mortality’s odds ratio (OR) for in-hospital serum ionized calcium derangements, compared with the persistently normal serum ionized calcium group, using logistic regression analysis. We adjusted OR for age, sex, race, principal diagnoses, comorbidities, estimated glomerular filtration rate, acute kidney injury, kidney replacement therapy, intensive care unit admissions, the number of in-hospital serum ionized calcium measurements, length of hospital stay, and admission serum ionized calcium. We considered a two-tailed p-value less than 0.05 as statistically significant. We performed all analyses using JMP statistical software (Version 10; SAS Institute Inc., Cary, NC, USA).

## 3. Results

### 3.1. Incidence of In-Hospital Hypocalcemia and Hypercalcemia

Figure 1 showed the study selection process. A total of 12,599 hospitalized patients were eligible for the analysis. The median number of in-hospital serum ionized calcium measurements was three (two to five). Among all patients, 6402 (51%) and 1544 (12%) developed new in-hospital hypocalcemia and hypercalcemia, respectively. Five thousand seven hundred and thirty-nine (46%) had persistently normal serum ionized calcium throughout the hospital stay, 5316 (42%) had in-hospital hypocalcemia only, 458 (4%) had in-hospital hypercalcemia only, and 1086 (8%) had both in-hospital hypocalcemia and hypercalcemia. Table 1 demonstrated clinical characteristics according to in-hospital serum ionized calcium groups.

The incidence of both in-hospital hypocalcemia and hypercalcemia was associated with younger age, female, non-Caucasian, admission for cardiovascular diseases, occurrence of acute kidney injury, need for ICU admission, greater number of serum ionized calcium measurements, and a longer length of hospital stay.

### 3.2. Association of In-Hospital Hypocalcemia and Hypercalcemia with Mortality

Mortality occurred in 3.1% of patients with in-hospital hypocalcemia, compared with 2.4% of patients without in-hospital hypocalcemia (*p* = 0.01). In-hospital hypocalcemia was significantly associated with increased in-hospital mortality, with an adjusted odds ratio of 1.29 (95% CI 1.01–1.66; *p* = 0.04) (Table 2).

The mortality occurred in 5.0% of patients with in-hospital hypercalcemia, compared with 2.4% in patients without in-hospital hypercalcemia. In-hospital hypercalcemia was significantly associated with increased in-hospital mortality with an adjusted odds ratio of 1.44 (95% CI 1.06–1.95); *p* = 0.02) (Table 2).

The mortality was 2.7% in patients with in-hospital hypocalcemia only, 5.2% in patients with in-hospital hypercalcemia only, and 4.9% in patients with both in-hospital hypocalcemia and hypercalcemia, compared with 2.1% in patients with persistently normal serum ionized calcium throughout the hospitalization. The in-hospital hypocalcemia only, in-hospital hypercalcemia only, and both in-hospital hypocalcemia and hypercalcemia groups were all significantly associated with increased in-hospital mortality, compared with the persistently normal serum ionized calcium group, with adjusted odds ratios of 1.28 (95% CI 1.01–1.64; *p* = 0.04), 1.64 (95% CI 1.02–2.68; *p* = 0.03), 1.73 (95% CI 1.14–2.62; *p* = 0.01), respectively (Table 2).

## 4. Discussion

Our study demonstrated that in-hospital serum ionized calcium derangements affected more than half of hospitalized patients (54%). Hypocalcemia occurred more often than hypercalcemia in the hospital. Moreover, we demonstrated for the first time that serum calcium disturbances in hospitalized patients are both common and associated with higher mortality.

In-hospital hypocalcemia and hypercalcemia were significantly associated with higher in-hospital mortality compared with persistently normal serum ionized calcium levels throughout hospitalization. This finding is in line that in-hospital mortality progressively increased with greater total calcium level changes, either in a decremental or incremental direction [12]. Our studies support the significance of calcium level fluctuations toward mortality. Hypercalcemia in the hospitalized setting is commonly seen in patients with cancers or endocrine/metabolic disorders [3,13,21]. An increase in calcium levels can cause renal vasoconstriction and nephrogenic diabetes insipidus, resulting in volume depletion and leading to a pre-renal AKI or exacerbating its severity [4]. AKI has been clearly linked with worse patient outcomes, including hospital and long-term mortality [22,23,24,25]. In addition, hypercalcemia is associated with acute neuronal injury [26] and acute congestive heart failure [27].

It has been observed that hypocalcemia is highly prevalent in critically ill patients and is associated with the severity of illness [2]. The underlying pathophysiology of this finding is not entirely understood. Some evidence suggests that resistance to the effects of PTH in the kidney and bone is involved [28,29,30]. Proinflammatory cytokines, namely tumor necrosis factor-α, have been linked to the severity of hypocalciuria and low concentrations of vitamin D metabolites in critically ill patients [29]. Moreover, hypomagnesemia, another common electrolyte complication in critically ill patients, could impair PTH secretion and thus has been suggested as a possible contributor to hypocalcemia [31,32]. Our study was adjusted for all possible confounders including comorbidities, and the association between in-hospital hypocalcemia and higher in-hospital mortality remains significant. Although the underlying mechanisms remain unclear, it is suggested that hypocalcemia can result in alteration of the myocardium action potential, leading to decreased cardiac contractility and acute pulmonary edema [17,33,34].

Our study has several limitations. First, this is a retrospective study from a single center and predominantly consisted of Caucasian patients. Hence, the generalizability of our findings may be limited. In addition, our cohort of hospitalized patients is highly selected. Serum ionized calcium is normally monitored in more critically ill patients, as supported by the finding that 69% of our cohort had ICU admission, or patients at high risk of serum calcium derangement. Thus, the incidence of serum calcium derangement reported in this observational study might over-estimate the actual incidence in general hospitalized patients. Second, we did not have data on vitamin D, PTH, serum pH, diet, or medications that might alter serum ionized calcium levels (oral or intravenous calcium or vitamin D supplements, diuretics), as well as the causes and timing of in-hospital dyscalcemia. Therefore, there may be unmeasured or residual confounders to our analysis. Nevertheless, the strengths of our study are worth mentioning. Firstly, we included a large cohort of 13,148 adult patients hospitalized with multiple principal diagnoses, and we adjusted for several potential confounders. Second, we excluded patients who had abnormal admission serum ionized calcium levels. This allowed us to ensure that we were explicitly investigating the incidence of in-hospital serum ionized calcium derangements and their impact on in-hospital mortality.

## 5. Conclusions

In conclusion, in-hospital serum ionized calcium derangements affected more than half of the hospitalized patients. In-hospital hypocalcemia and hypercalcemia were both significantly associated with increased in-hospital mortality.

## Figures and Tables

**Figure 1 medicines-07-00070-f001:**
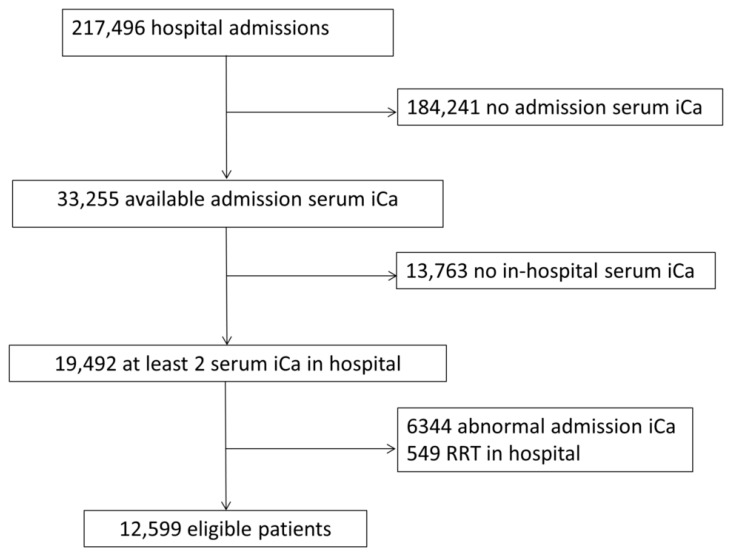
Study selection flow.

**Table 1 medicines-07-00070-t001:** Clinical characteristics of study patients.

Variables	All	Serum Ionized Calcium during Hospitalization				
		Normal	Hypocalcemia Only	Hypercalcemia Only	Both Hypo- and Hypercalcemia	*p*-Value
N	12,599	5739	5316	458	1086	
Age (year)	63 ± 17	63 ± 17	62 ± 16	66 ± 16	62 ± 16	<0.001
Male sex	7250 (58)	3181 (55)	3265 (61)	232 (51)	572 (53)	<0.001
Caucasian	11,584 (92)	5294 (92)	4902 (92)	425 (93)	963 (89)	0.001
Principal diagnosis						<0.001
- Cardiovascular	4281 (34)	1138 (20)	2321 (44)	141 (31)	681 (63)	
- Hematology/oncology	2570 (20)	1363 (24)	1023 (19)	79 (17)	105 (10)	
- Infectious disease	365 (3)	176 (3)	137 (3)	27 (6)	25 (2)	
- Endocrine/metabolic	252 (2)	149 (3)	80 (2)	16 (3)	7 (1)	
- Respiratory	548 (4)	346 (6)	151 (3)	28 (6)	23 (2)	
- Gastrointestinal	1041 (8)	611 (11)	338 (6)	46 (10)	46 (4)	
- Genitourinary	271 (2)	156 (3)	81 (2)	22 (5)	12 (1)	
- Injury and poisoning	1733 (14)	961 (17)	637 (12)	55 (12)	80 (7)	
- Other	1538 (12)	839 (15)	548 (10)	44 (10)	107 (10)	
Comorbidity						
- Coronary artery disease	2940 (23)	1307 (23)	1241 (23)	127 (28)	265 (24)	0.08
- Congestive heart failure	1057 (8)	443 (8)	444 (8)	46 (10)	124 (11)	<0.001
- Peripheral artery disease	555 (4)	265 (5)	211 (4)	28 (6)	51 (5)	0.09
- Stroke	1032 (8)	498 (9)	376 (7)	46 (10)	112 (10)	<0.001
- Diabetes mellitus	2755 (22)	1333 (23)	1094 (21)	114 (25)	214 (20)	0.001
- COPD	1383 (11)	672 (12)	525 (10)	65 (14)	121 (11)	0.002
- Cirrhosis	287 (2)	127 (2)	113 (2)	12 (3)	35 (3)	0.15
Charlson Comorbidity Score	2.0 ± 2.4	2.2 ± 2.5	1.8 ± 2.3	2.6 ± 2.7	1.6 ± 2.0	<0.001
eGFR (ml/min/1.73 m^2^)	77 ± 28	77 ± 29	77 ± 26	65 ± 32	76 ± 25	<0.001
Acute kidney injury	2611 (21)	869 (15)	1215 (23)	140 (31)	387 (36)	<0.001
ICU admission	8691 (69)	2947 (51)	4369 (82)	314 (69)	1061 (98)	<0.001
Number of serum ionized calcium measurement in hospital	3 (2–5)	2 (2–3)	4 (3–6)	4 (3–7)	5 (4–9)	<0.001
Length of hospital stay (day)	6 (4–9)	5 (3–8)	6 (4–10)	7 (5–13)	7 (5–11)	<0.001
Admission serum ionized calcium (mg/dL)	4.80 (4.70–4.93)	4.85 (4.73–5.00)	4.76 (4.68–4.86)	4.99 (4.81–5.14)	4.80 (4.70–4.90)	<0.001
Lowest serum ionized calcium (mg/dL)	4.57 (4.23–4.75)	4.75 (4.66–4.85)	4.28 (4.04–4.45)	4.85 (4.70–5.01)	4.11 (3.94–4.30)	<0.001
Highest serum ionized calcium (mg/dL)	4.98 (4.82–5.20)	4.97 (4.85–5.10)	4.90 (4.77–5.09)	5.60 (5.46–5.81)	5.73 (5.54–6.20)	<0.001

Continuous data are presented as mean ± SD or median (IQR); categorical data are presented as count (%).

**Table 2 medicines-07-00070-t002:** Association between in-hospital serum ionized calcium derangements and in-hospital mortality.

Serum Ionized Calcium during Hospitalization	N	In-Hospital Mortality	Univariable Analysis		Multivariable Analysis	
			OR (95% CI)	*p*	Adjusted OR (95% CI)	*p*
In-hospital hypocalcemia						
No	6197	147 (2.4)	1 (ref)	-	1 (ref)	-
Yes	6402	198 (3.09)	1.31 (1.06–1.63)	0.01	1.29 (1.01–1.66)	0.04
In-hospital hypercalcemia						
No	11,055	268 (2.4)	1 (ref)	-	1 (ref)	-
Yes	1544	77 (5.0)	2.11 (1.63–2.74)	<0.001	1.44 (1.06–1.95)	0.02
Groups						
Normal	5739	123 (2.1)	1 (ref)	-	1 (ref)	-
Hypocalcemia only	5316	145 (2.7)	1.28 (1.01–1.63)	0.04	1.28 (1.01–1.64)	0.04
Hypercalcemia only	458	24 (5.2)	2.52 (1.61–3.95)	<0.001	1.64 (1.02–2.68)	0.03
Both hypo- and hypercalcemia	1086	53 (4.9)	2.34 (1.69–3.25)	<0.001	1.73 (1.14–2.62)	0.01

Adjusted for age, sex, race, principal diagnosis, Charlson comorbidity score, coronary artery disease, congestive heart failure, peripheral vascular disease, stroke, diabetes mellitus, chronic obstructive pulmonary disease, cirrhosis, eGFR, acute kidney injury, ICU admission, the number of serum ionized calcium measurements, length of hospital stay, admission serum ionized calcium.

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
