# Peer review of "Hospital-Acquired Serum Ionized Calcium Derangements and Their Associations with In-Hospital Mortality"

_medicines, 2020, doi:10.3390/medicines7110070_

Round 1

Reviewer 1 Report

  1. Numerical variables are reported as mean±SD or median(IQR) but the method is not presented in the statistical analysis. It must be specified what normality test was employed.
  2. Table 1 - where the data had non-Gaussian distribution (eg. Number of calcium measurement, hospitalisation days) the ANOVA test is not appropriate and it should be employ a non-parametric test (eg. Kruskal-Wallis)
  3. Table 1 - lines seem to be offset (especially the p-value in the line with principal diagnosis), it must be aligned properly. 

Overall it is a good article, the database is big, the results are clearly presented, the statistical analysis is done well, and the study limitations are very well pointed out.

It will be nice to see a more in-depth analysis of both hypo- and hypercalcemia group and to assesses the predictive factors since this group presented the highest risk for mortality.

Author Response

Response to Reviewer#1

Comment #1

Numerical variables are reported as mean±SD or median(IQR) but the method is not presented in the statistical analysis. It must be specified what normality test was employed.

Response: The following statements have been added to the statistical analysis section.

“We tested the normality of continuous variables using Shapiro-Wilk test. We presented continuous variables as mean ± standard deviation (SD) for normally-distributed data, or median (interquartile rate [IQR]) for skewed data. We compared continuous variables using analysis of variance for normally-distributed data, or Kruskal-Wallis test for skewed data.”

Comment #2

Table 1 - where the data had non-Gaussian distribution (eg. Number of calcium measurement, hospitalisation days) the ANOVA test is not appropriate and it should be employ a non-parametric test (eg. Kruskal-Wallis)

Response:  The following statements have been added to the statistical analysis section.

“We compared continuous variables using analysis of variance for normally-distributed data, or Kruskal-Wallis test for skewed data.”

We updated the analysis using non-parametric test for non-normally distributed data.

Comment #3

Table 1 - lines seem to be offset (especially the p-value in the line with principal diagnosis), it must be aligned properly.

Response: This error has been corrected.

Comment #4

Overall it is a good article, the database is big, the results are clearly presented, the statistical analysis is done well, and the study limitations are very well pointed out.

It will be nice to see a more in-depth analysis of both hypo- and hypercalcemia group and to assesses the predictive factors since this group presented the highest risk for mortality.

Response: The following statements have been added to the result.

The incidence of both in-hospital hypocalcemia and hypercalcemia were associated with younger age, female, non-Caucasian, admission for cardiovascular diseases, occurrence of acute kidney injury, need for ICU admission, more number of serum ionized calcium measurements, and longer length of hospital stay.

We greatly appreciated the reviewer’s time and comments to improve our manuscript. The manuscript has been improved considerably by the suggested revisions.

Reviewer 2 Report

This article studies the relationship of development of hypo or hypercalcemia while in the hospital with mortality. The article is of sound design and conclusions are appropriate. I definitely think the article is worthy of publication. I see this article not changing clinical practice at this time but it is a step towards studying whether treating calcium abnormalities will affect prognosis. I only have minor revision requests:

  1. Is it possible to analyze how many of the patients in each group were receiving calcium or vitamin D?
  2. Do the authors have data on what was the cause of the hypercalcemia on the patients who developed it?
  3. On Table 1. Thirty nine patients received renal replacement therapy while 53 patients had ESRD! Is this because the rest of the patients had kidney transplant?
  4. We often think of kidney disease as leading to hypocalcemia not hypercalcemia. However according to Table 1 the eGFR was the lowest in the patients with hypercalcemia. How do the authors explain this paradox?
  5. Sentences should not begin with a numerical value. rather than begin a sentence with "5,495", state "Five thousand four hundred ninety-five."
  6. It would also be useful to get an idea of serum CO2 concentrations in each group. As mildly low serum ionized calcium concentration may often be due to metabolic acidosis which indicate a worse prognosis. 
  7. It would also be useful to know in patients with AKI, if the hypo or hypercalcemia happened before or after the loss of renal function. 

Author Response

Response to Reviewer#2

This article studies the relationship of development of hypo or hypercalcemia while in the hospital with mortality. The article is of sound design and conclusions are appropriate. I definitely think the article is worthy of publication. I see this article not changing clinical practice at this time but it is a step towards studying whether treating calcium abnormalities will affect prognosis. I only have minor revision requests.

Response: We thank you for reviewing our manuscript and for your critical evaluation.

Comment #1

Is it possible to analyze how many of the patients in each group were receiving calcium or vitamin D?

Response: Our database did not contain medication administration in hospital. The following statements have been added to the limitation.

“We did not have data on vitamin D, PTH, serum pH, diet, or medications that might alter serum ionized calcium levels (oral or intravenous calcium or vitamin D supplements, diuretics), as well as the causes and timing of in-hospital dyscalcemia. Therefore, there may be unmeasured or residual confounders to our analysis.”

Comment #2

Do the authors have data on what was the cause of the hypercalcemia on the patients who developed it?

Response: The causes of in-hospital hypocalcemia or hypercalcemia are often multifactorial and complex. It would require comprehensive medical record review to determine the causes of dyscalcemia. Despite comprehensive medical record review, the causes of dyscalcemia are sometimes unable to be determined. The following statements have been added to the limitation.

“We did not have data on vitamin D, PTH, serum pH, diet, or medications that might alter serum ionized calcium levels (oral or intravenous calcium or vitamin D supplements, diuretics), as well as the causes and timing of in-hospital dyscalcemia. Therefore, there may be unmeasured or residual confounders to our analysis.”

Comment #3

On Table 1. Thirty nine patients received renal replacement therapy while 53 patients had ESRD! Is this because the rest of the patients had kidney transplant?

Response: We apologized for this error. We initially included kidney transplant patients in end-stage renal disease patients. As suggested by another reviewer, we excluded patients who received renal replacement therapy in hospital from our analysis as renal replacement therapy can significantly altered serum ionized calcium.

Comment #4

We often think of kidney disease as leading to hypocalcemia not hypercalcemia. However according to Table 1 the eGFR was the lowest in the patients with hypercalcemia. How do the authors explain this paradox?

Response: The reviewer has great knowledges in Nephrology and we agree with this point that patients with chronic kidney disease usually develop hypocalcemia, especially when patients reach chronic kidney disease stage 4. Although patients who developed hospital acquired hypercalcemia had lower baseline eGFR than other groups, these still were in the ranges that were less likely to develop hypocalcemia from CKD. Furthermore, our studies assessed hospital acquired derangements, and hypercalcemia development was the outcome during hospitalization and was not at baseline at the same time of baseline characteristics. We appreciate the reviewer’s great point.

Comment #5

Sentences should not begin with a numerical value. rather than begin a sentence with "5,495", state "Five thousand four hundred ninety-five."

Response: This change has been made as suggested.

Comment #6

It would also be useful to get an idea of serum CO2 concentrations in each group. As mildly low serum ionized calcium concentration may often be due to metabolic acidosis which indicate a worse prognosis.

Response: We agreed that acid-base disturbance can change serum ionized calcium as the reviewer pointed out. However, it is serum pH, not serum bicarbonate, that indeed alters serum ionized calcium. As serum pH was missing in majority of hospitalized patients, we did not report serum pH in our study. The following statements have been added to the limitation.

“ We did not have data on vitamin D, PTH, serum pH, diet, or medications that might alter serum ionized calcium levels (oral or intravenous calcium or vitamin D supplements, diuretics), as well as the causes of in-hospital dyscalcemia. Therefore, there may be unmeasured or residual confounders to our analysis”

Comment #7

It would also be useful to know in patients with AKI, if the hypo or hypercalcemia happened before or after the loss of renal function.

Response:  As both serum ionized calcium and kidney function can fluctuate throughout the hospitalization, it is difficult to adjudicate the timing of dyscalcemia in relation to kidney dysfunction. The following statements have been added to the limitation.

“We did not have data on vitamin D, PTH, serum pH, diet, or medications that might alter serum ionized calcium levels (oral or intravenous calcium or vitamin D supplements, diuretics), as well as the causes and timing of in-hospital dyscalcemia. Therefore, there may be unmeasured or residual confounders to our analysis.”

We greatly appreciated the reviewer’s time and comments to improve our manuscript. The manuscript has been improved considerably by the suggested revisions.

Reviewer 3 Report

The author address the impact of in hospital hypo or hyper calcemia based on the determination of ionized Ca.

The measurement of ionized Ca measurement is quite complex and nothing is mentioned about the preanalytical and analytical management of blood samples

We have no data on the type of blood sample used (venous, arterial or capillary). In the same way, nothing is mentioned about blood sample conservation (4°C or analyzed in total blood sample using PH measurement), the type of blood tube

We do not know if ionized calcium was corrected with pH

Furthermore, in 2000, IFCC (international Federation of Clinical Chemistry) defined a reference method for the determination of ionized ca in plasma. Do we know if the analysis of blood samples was modified during the study period (from 2009 to 2013) (PMID 11205698; DOI: 10.1515/CCLM.2000.206)

Results

Is it possible to have the total number of patients hospitalized during the study period (How many patients have 3 or more measurement of ionized Ca in their inpatient population during the same period)?

It would be of interested to have data on the distribution of ionized Ca as described in previous publication (ref 12)

The impact of renal replacement therapy on blood calcium levels can be due to the anticoagulation used in these patients (citrate anticoagulation). This represents a bias in the interpretation of the data and I would be interested in statistical analysis (ANOVA) without these patients.

In Table 1.How can they explain the line “highest serum concentration of ionized Ca” in the hyperCa group at 5.79 +/- 0.58 (this result is not consistent with hyperCa as 68.3% would have a calcemia between 5.21 and 6.37mg/dl while hyperCa is defined as greater than 5.4mg/dl). This result may be consistent with hypo and hyperCa

In adjusted models, some variable are strongly inter-correlated (Acute kidney injury and renal remplacement therapy, ICU admission and RRT…)

Discussion

How can they explain their really high proportion of abnormal Ca in their population? In the literature, outside of their publications that always use the same cohort (ref 12, 13, 19), the proportion of dyscalcemia is described between 15 and 30%. What is the particularity of their population?

Author Response

Response to Reviewer#3

Comment #1

The author address the impact of in hospital hypo or hypercalcemia based on the determination of ionized Ca.

The measurement of ionized Ca measurement is quite complex and nothing is mentioned about the preanalytical and analytical management of blood samples

We have no data on the type of blood sample used (venous, arterial or capillary). In the same way, nothing is mentioned about blood sample conservation (4°C or analyzed in total blood sample using PH measurement), the type of blood tube.

We do not know if ionized calcium was corrected with pH.

Furthermore, in 2000, IFCC (international Federation of Clinical Chemistry) defined a reference method for the determination of ionized ca in plasma. Do we know if the analysis of blood samples was modified during the study period (from 2009 to 2013) (PMID 11205698; DOI: 10.1515/CCLM.2000.206)

Response: The following statements have been added to the method section.

“All serum ionized calcium values were analyzed from venous blood samples collected in serum separating tubes or serum gel tubes using ion-selective electrode method throughout the study period. Serum ionized calcium was adjusted to pH 7.40 to account for changes in specimen pH that may occur during transport.”

Comment #2

Is it possible to have the total number of patients hospitalized during the study period (How many patients have 3 or more measurement of ionized Ca in their inpatient population during the same period)?

Response:  We added figure 1 to describe the study selection flow (attached PDF)

Comment #3

It would be of interested to have data on the distribution of ionized Ca as described in previous publication (ref 12)

Response: Data on the distribution of admission, the lowest, and the highest serum ionized calcium, summarized as median with interquartile range, were shown in Table 1.

Comment #4

The impact of renal replacement therapy on blood calcium levels can be due to the anticoagulation used in these patients (citrate anticoagulation). This represents a bias in the interpretation of the data and I would be interested in statistical analysis (ANOVA) without these patients.

Response: We agreed with the reviewer that the inclusion of patients who required renal replacement therapy in the analysis can considerably bias the interpretation of data. Therefore, we excluded patients who received renal replacement therapy from the analysis. The results have been updated throughout the manuscript.

Comment #5

In Table 1.How can they explain the line “highest serum concentration of ionized Ca” in the hyperCa group at 5.79 +/- 0.58 (this result is not consistent with hyperCa as 68.3% would have a calcemia between 5.21 and 6.37mg/dl while hyperCa is defined as greater than 5.4mg/dl). This result may be consistent with hypo and hyperCa

Response: The below figure showed the distribution, percentiles, and mean ± standard deviation of the highest serum ionized calcium in in-hospital hypercalcemia only group. (Figure in attached PDF)

The minimum value of the highest serum ionized calcium in this group was 5.41. Therefore, all patients in this group met the definition of hypercalcemia.

The below figure showed the distribution, percentiles, and mean ± standard deviation of the lowest serum ionized calcium in in-hospital hypercalcemia only group (Figure in Attached PDF)

The minimum value of the lowest serum ionized calcium in this group was 4.6. Therefore, all patients in this group did not meet the definition of hypocalcemia.

As the data of serum ionized calcium is skewed, mean ± standard deviation does not well represent the data and cause confusion. We summarized the data of serum ionized calcium using median with interquartile range instead.

Of note, all data in table 1 was updated as we excluded patients with in-hospital renal replacement therapy as  explained in comment#4.

Comment #6

In adjusted models, some variable are strongly inter-correlated (Acute kidney injury and renal replacement therapy, ICU admission and RRT…)

Response: As explained in comment#4, we excluded patients with in-hospital renal replacement therapy. Therefore, we did not adjust for renal replacement therapy in multivariable models.

Comment #7

How can they explain their really high proportion of abnormal Ca in their population? In the literature, outside of their publications that always use the same cohort (ref 12, 13, 19), the proportion of dyscalcemia is described between 15 and 30%. What is the particularity of their population?

Response: The following statements have been added to the limitation to acknowledge the possible over-estimation of incidence of dyscalcemia in our study.

“In addition, our cohort of hospitalized patients is highly selected. Serum ionized calcium is normally monitored in more critically ill patients, as supported by the finding that 69% of our cohort had ICU admission, or patients at high risk of serum calcium derangement. Thus, the incidence of serum calcium derangement reported in this observational study might over-estimate the actual incidence in general hospitalized patients.”

We greatly appreciated the reviewer’s time and comments to improve our manuscript. The manuscript has been improved considerably by the suggested revisions.

Round 2

Reviewer 3 Report

Correction of the article is appropriate.